# Using Reinforcement Learning to Investigate Neural Dynamics During Motor Learning

## Abstract

Recent work characterized shifts in preparatory activity of the motor cortex during motor learning. The specific geometry of the shifts during learning, washout, and relearning blocks was hypothesized to implement the acquisition, retention, and retrieval of motor memories. We sought train recurrent neural network (RNN) models that could be used to study these motor learning phenomena. We built an environment for a curl field (CF) motor learning task and trained RNNs with reinforcement learning (RL) with novel regularization terms to perform behaviorally realistic reaching trajectories over the course of learning. Our choice of RL rather than supervised learning was motivated by the idea that motor adaptation to a novel environment, in the absence of demonstrations, is a process of reoptimization. We find these models, despite lack of supervision, reproduce many behavioral findings from human and monkey CF adaptation experiments. Relearning is faster than initial learning, indicating formation of motor memories. Optimal reaches under a CF are not straight, but rather curved, which is optimal and has been observed in humans and macaques. These models also captured key neurophysiological findings. We found that the model's preparatory activity existed in a force-predictive subspace that remained stable across learning, washout, and relearning. Additionally, preparatory activity shifted uniformly, independently of the distance to the CF trained target. Finally, we found that the washout shift became more orthogonal to the learning shift, and hence more brain-like, when the RNNs were pretrained to have prior experience with CF dynamics. We argue the increased fit to neurophysiological recordings is driven by more generalizable and structured dynamical motifs in the model with more prior experience. This suggests that the near-orthogonality of learning-washout neural geometry underlying motor memory may be influenced by structured dynamical motifs in the motor cortex circuitry developed from prior experience. Together, our work takes a step towards elucidating the factors that support motor memory, acquisition, retention, and retrieval during motor learning.

## 1 Introduction

How does the brain learn? One approach to this question is to view the brain under the framework of deep learning (Richards et al., 2019). The authors argue that with the tremendous success of deep learning in many tasks, it may benefit systems neuroscience to view the brain through the lens of deep learning, seeking to identify the brain's objective functions, learning rules, and architectures, in the context of the set of tasks it is optimized for. Parallel with the growth of deep learning, neuroscience has experienced exponential, Moore's-law-like, growth in the number of simultaneously recorded neurons as animals perform a range of tasks (Urai et al., 2022), creating a need for modeling and theoretical methods that scale with recording growth. One consequence of this parallel growth in deep learning and systems neuroscience is growing evidence that artificial neural networks reproduce key features of neural activity recorded from various brain areas in animals across a broad array of tasks (Sussillo et al., 2015; Mante et al., 2013; Rajan et al., 2016). In systems neuroscience, these trained or designed recurrent neural networks (RNNs) are then analyzed to explain either the "why" or "how" of various cognitive or motor neural computations. These models have been successful in elucidating novel mechanistic insight in many tasks, including decision-making (Mante et al., 2013; Barak et al., 2013), movement generation (Sussillo et al., 2015), timing (Rajan

et al., 2016; Wang et al., 2018), working memory (Orhan & Ma, 2019) and exploration (Banino et al., 2018).

In this paper, we don't seek to characterize or hypothesize how a biological network expertly performs a cognitive, motor, or navigational task, but rather seek to match underlying neural activity implementing the task *over the course of learning* and hypothesize how ecologically relevant prior experience influences neural activity underpinning learning. Critically, motor learning has important differences compared to prior studies on decision-making, movement generation, working memory, and timing. These studies have been primarily concerned with the dynamical mechanisms within a static network that implement these behaviors, and aimed to explain neural computations at a snapshot in time, instead of how the dynamical mechanisms are organized across learning.

The task we model is a motor adaptation experiment conducted with macaques (Sun et al., 2022). This study characterized the systematic changes in neural activity underlying the learning, unlearning ("washout"), and relearning of a curl field perturbation on delayed reaches to targets. It also uncovered structured relationships between the activity changes underlying the acquisition of different trained target configurations. The highly systematic geometric relationship between these changes were hypothesized to implement motor memory in the brain. These results will serve as a detailed guide for evaluating the behavioral and neural characteristics of RNNs trained with RL.

To create a model of, and generate a hypothesis regarding the learning process in (Sun et al., 2022), we argue that it is natural and important that reinforcement learning (RL) is used to train the networks. In Sun et al. (2022), the monkey learns to reach under a curl field in the absence of demonstrations. Not knowing, a priori, the optimal behavior makes supervised learning and imitation learning an unnatural choice. Since, the monkey is instead rewarded with a small dose of fruit juice at the end of a successful reach to a target, it is more natural to train our RNN model of motor learning with RL (Botvinick et al., 2020) Also, in similar motor learning tasks to the one we seek to model (Sun et al., 2022), the learning process is driven by performance reoptimization (Izawa et al., 2008). We therefore use PPO (Schulman et al.) to train our RNN models and evaluate whether the model exhibits similar behavioral and neural adaptation to animals.

Our contributions are as follows. 1) Artificial models can generate physically unrealistic accelerations, resulting in non-physiological behavior. We therefore develop action distribution regularization terms that lead to realistic accelerations during reaches. 2) We build an artificial environment that replicates the curl field adaptation experiment in Sun et al. (2022), and train an RNN to learn this task using deep RL. We show the trained RNN replicates key behavioral features of motor learning. 3) We show that our model reproduces several key neurophysiological features of motor adaptation, including a stable, reused, force-predictive subspace for preparatory activity, and uniform shift of preparatory states. 4) Finally, we show that providing the model with prior experience reproduces additional neurophysiological features such as near-orthogonality of learning and washout shifts, demonstrating that matching the task set between artificial and animal models (Richards et al., 2019) can improve neural similarity. We believe this study demonstrates that RNNs trained with RL are a useful source of hypothesis generation that can support progress in the theoretical understanding of *why* systematic brain activity emerges during motor adaptation, and how motor memory is implemented.

## 2 METHODS

We first describe an environment[1] of the delayed reach motor adaptation task from Sun et al. (2022). We then detail how we modify the PPO training objective to include behavioral regularization terms, and finally pretrain the RNNs to learn to reach without CF perturbations, then proceed through the stages of learning (a CF perturbation on one target), washout, and relearning.

### 2.1 CENTER-OUT-AND-BACK TASK WITH CURL FIELD

To study the systematic neural activity changes that occur during motor adaptation, Sun et al. (2022) designed an curl field learning task. We replicate this task here (Fig. 1a). The agent in the environment (RNN) learns a policy $\pi_\theta(\cdot|\mathbf{s})$ that outputs a distribution over reach acceleration in a 2D plane

---

[1]We implemented this as a custom environment using OpenAI's Gym.

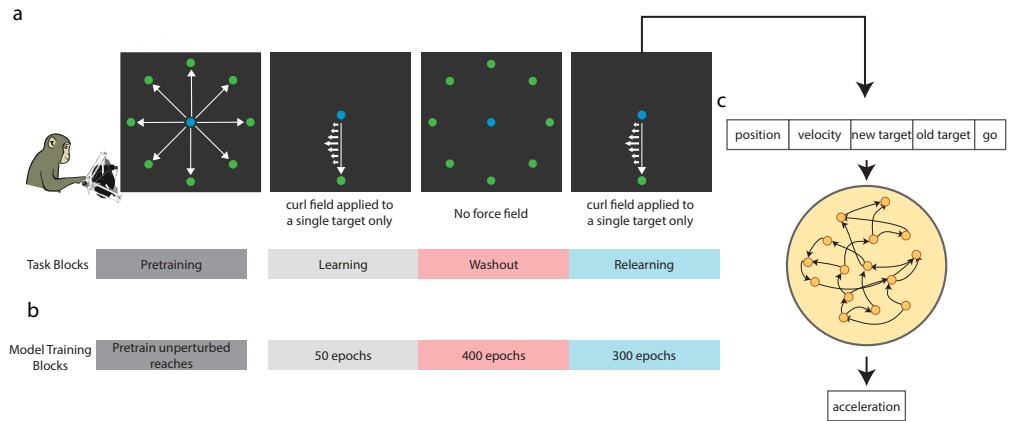

Figure 1: **(a)** Task blocks of our curl field adaptation task (Sun et al., 2022). In the case of the monkey, "pretraining" is not required, as monkeys naturally learn to reach skillfully throughout development and life. **(b)** For our models, on the other hand, we performed a lengthy period of pretraining from scratch with a curriculum. **(c)** Our model is a simple LSTM processing the 9D environment state as input.

given the environment state, **s**. This acceleration controls the position of a cursor in the plane. The center-out-and-back task comprises are 8 radial targets (Fig. 1a), located the same radial distance (12 cm) from the center of the 2D environment. There is also one center target at (0,0). The agent must perform a delayed reach task, with or without the curl field (CF), in this environment. This task is described further below.

**Curl field:**    Normally, the cursor moves naturally based on the model's output acceleration. When a CF is applied, the CF force induces an additional acceleration term added to the model's output acceleration before updating the environment,

$$\begin{bmatrix} A_x \\ A_y \end{bmatrix} = k \begin{bmatrix} 0 & -1 \\ 1 & 0 \end{bmatrix} \begin{bmatrix} V_x \\ V_y \end{bmatrix}. \tag{1}$$

This perturbation to the acceleration is proportional to the velocity and is rotated by 90 degrees. An example curl field with respect to the reach trajectory is shown in Fig. 2f.

**Environment state and policy input:**    The RNN policy receives as input the environment state at time $t$, denoted $\mathbf{s}_t$. The environment state is 9D, comprising the 2-dimensional cursor position, cursor velocity, start target position, goal target position, and the go cue (1D) at every time step $t$. The RNN inputs were therefore also 9D, as in Fig.1c.

**Time discretization:**    Time is discretized into windows of 50 ms, so that each characteristic period of a reach is on the order of 5 to 60 timesteps.

**Task structure and rewards: pretraining and washout**    At the beginning of a trial (environment episode), the cursor position is initialized to be at the start target with zero velocity. The goal target appears, modifying the environment state. In the delayed reach task, the agent must acquire the goal target but can only reach from the start target after a go cue is given. To hold the start target, the RNN agent must keep the cursor position within some tolerance window of the start target for a random delay period ranging from 0 to 500 ms. If the cursor leaves this window, the episode terminates with zero reward and another episode then begins, with the cursor position reset to the start target. If the agent keeps the cursor over the start target for the duration of the delay period, then the environment state go cue transitions from 0 to 1, indicating the agent can make a reach to the goal target. A trial is successful if the agent moves the cursor to the goal target and stays within an acceptance window for 500 ms. The environment returns a reward of 1 for a successful trial. Following a successful trial, the current goal target becomes the new start target, and a new goal target is either set to the center target, if a center-out reach was just performed; Or, a random target among the 8

radial targets, if a center-in reach was just performed. If the agent does not successfully acquire the target within 3 seconds, the episode (trial) terminates with zero reward. During pretraining, the RNN policy receives the curl field strength, $k$, but during washout it does not receive $k$ as an input.

**Task structure and rewards: learning and relearning**   This environment is the same as the pretraining and washout environment, except there is only one target (see Fig.1a), following the design of Sun et al. (2022). The agent therefore performs out and back reaches between the center target and one radial target with a CF applied. Throughout this paper, the CF is a constant curl field Eq. 1 with coefficient $k = 0.4$. This coefficient, $k$, is not observed by the RNN agent during CF learning and relearning.

## 2.2   MODEL ARCHITECTURE AND TRAINING

We implemented our agent policy $\pi_\theta$ with RNNs and trained them to maximize rewards in the environment using proximal policy optimization (PPO) (Schulman et al.).

**Model architecture:**   PPO is an actor-critic method, where the actor is the agent policy, $\pi_\theta$, and the critic is the value function, which is used to compute clipped advantages. We used a shared actor and critic network, which is a single layer LSTM with a hidden and cell size of 256. The actor and critic are two separate linear readouts of the LSTM hidden state. For the actor, a linear layer after the LSTM outputs the mean and logarithmic standard deviation of Gaussian distributions for the cursor's $x$ and $y$ acceleration. For the critic, a linear layer after the LSTM outputs the state value. As discussed in Section 2.1, the model input is the 9D environment state.

We chose to use a LSTM for scientific and practical reasons. Primarily, we use a recurrent architecture as these architectures have reproduced many key features of neural activity in a range of motor (Sussillo et al., 2015) and non-motor (Mante et al., 2013; Rajan et al., 2016; Orhan & Ma, 2019) tasks when trained in a purely task driven manner, sometimes with minimal explicit constraints to improve match to neural activity. We emphasize that we chose a recurrent architecture not due to performance considerations, but in order to investigate the unit population activity across stages of learning in the motor adapation task from Sun et al. (2022). We use LSTMs instead of "vanilla" RNNs, because we found that "vanilla" RNNs did not converge reliably in training.

**Training: policy regularizations to ensure realistic reaching trajectories**   We build on top of the standard PPO algorithm (see Schulman et al. for additional details). The standard PPO objective is

$$L_t^{PPO}(\theta) = \hat{\mathbb{E}}_t \left[ \min \left( r_t(\theta) \hat{A}_t, \ \text{clip}(r_t(\theta), 1 - \epsilon, 1 + \epsilon) \hat{A}_t \right) - c_1 L_t^{VF}(\theta) + c_2 S[\pi_\theta](s_t) \right] \quad (2)$$

When training our deep RL model with this unmodified PPO objective, we observed model policies produced reach accelerations that were apparently increasing without bound (Fig.2a), violating biophysical constraints in physical arms. For our application, it is important that the behavioral output of the model matches experiments before we attempt to analyze the model activity and make comparisons to neurophysiological recordings. We incorporated two constraints in order to combat overly large accelerations. The first constraint is a zeroness constraint that penalizes the KL divergence between the policy and a zero mean unit variance Gaussiain. This penalizes the model for excessively high accelerations. The second constraint is a smoothness constraint, that penalizes the KL divergence between the policy at one time with the policy at the next time step. This encouraged the accelerations to not change to abruptly. This is motivated by the finding that human hand paths are solutions to an optimal control problem that optimize a composition of mechanical energy expenditure and joint smoothness, which loosely correspond to our zeroness and smoothness constraints (Berret et al., 2011). Our final training objective then became

$$L_t(\theta) = L_t^{PPO}(\theta) - \hat{\mathbb{E}}_t \left[ \alpha_{smoothness} KL(\pi_\theta(\cdot|s_t)||\pi_\theta(\cdot|s_{t+1})) - \alpha_{zeroness} KL(\pi_\theta(\cdot|s_t)||\mathcal{N}(0,1)) \right]$$
$$(3)$$

We optimize this objective using Adam (Kingma & Ba, 2017), with a learning rate of $3 \times 10^{-5}$, epsilon= $1 \times 10^{-5}$, and otherwise using default values [2]. We use the term "epoch" in an overloaded way, to describe one cycle of trajectory collection for 20,000 timesteps, and 4 "optimization epochs" (Schulman et al.).

**Curriculum Learning (during pretraining):**  When the monkey began experiments, it could, due to prior experience, already reach along straight trajectories towards radial targets, without a curl field perturbation. We likewise pretrain the model to also be able to perform straight reaches in an unperturbed environment (no curl field). We designed a curriculum to combat the sparse nature of the naïve reward, where a reward of 1 is only given when a target is acquired. We avoided reward shaping, as this can unnaturally influence the learning process. The curriculum begins with an acceptance (for the goal target) and tolerance (for the start target) window with side length 12 cm (area: $144 \text{ cm}^2$). This is a very large initial window, as it is the distance from the center target to a radial target. We defined that when the model reaches with $> 95\%$ success accuracy, the acceptance window is reduced by 10%. The curriculum continues until the acceptance window is of area 1 cm$^2$. We observed that pretraining always converged under this curriculum.

**Compute Usage**  Training took place entirely on a AMD Ryzen Threadripper 3960X 24-Core CPU. Pretraining and a full adaptation experiment for 4 seeds took approximately 40 hours.[3]

## 3 RESULTS

We train the RNN model with 4 seeds. The model was trained through curriculum learning to perform unperturbed delayed reaches to 8 center-out targets.

### 3.1 POLICY REGULARIZATION INDUCES REALISTIC REACH PROFILES

We first found that the policy regularization we introduced in Section 2.2 resulted in realistic reach profiles. Models learned to hold still until go cue presentation. At go cue presentation, models generated a modest acceleration to initiate the reach, then subsequently decelerated. Models then held the cursor still over the goal target. Example trajectories during delayed reaches to all 8 targets are shown in Fig. 2d.

We performed an ablation of the policy regularization terms in the no-CF pretraining condition. As Fig. 2a shows, without regularization, peak acceleration increases nearly linearly with training beginning from when the RNN model successfully learns the curriculum, reflecting the reward focused optimization. With the smoothness constraint, the peak acceleration drops substantially. With the zeroness constraint or both constraints, max acceleration saturates at around 2 m/s$^2$ at or before completion of the curriculum. The impact of these regularization terms on the acceleration profile is shown in Fig. 2b. Note the qualitative similarity between the RNN's acceleration profiles with the monkey's in Fig. 2c. With this, we have shown that regularization terms can control model behavior without direct supervision. These terms can be used to impose more biologically plausible behavior on the RNN models when modeling motor tasks. For the rest of model training, the zeroness and smoothness coefficients were both set to $0.001$. This was selected empirically for approximate behavioral match, but can be fine tuned for more exact match of the acceleration profile.

### 3.2 RNNs TRAINED WITH RL REPLICATE KEY BEHAVIORAL CHARACTERISTICS

To our knowledge, no prior work has demonstrated that RNN models trained with RL achieve similar reaching behavior to macaques. We show that the key behavioral features of macaques during a motor adaptation task (Sun, 2022) are replicated by our model.

**RL models successfully learn to reach through CF**  We found that our RNN policy was able to, with RL, learn to reach under the curl field in 100 epochs (Fig.2g, black solid line). Due to the action

---

[2]We trained our models with Stable-Baselines1, which is implemented on Tensorflow 1. To implement the smoothness constraint during training, we needed to create a copy of the policy network in the dataflow graph using TF1's variable scoping and "reuse" option.

[3]The experiments reported in this paper took approximately 10 CPU Days.

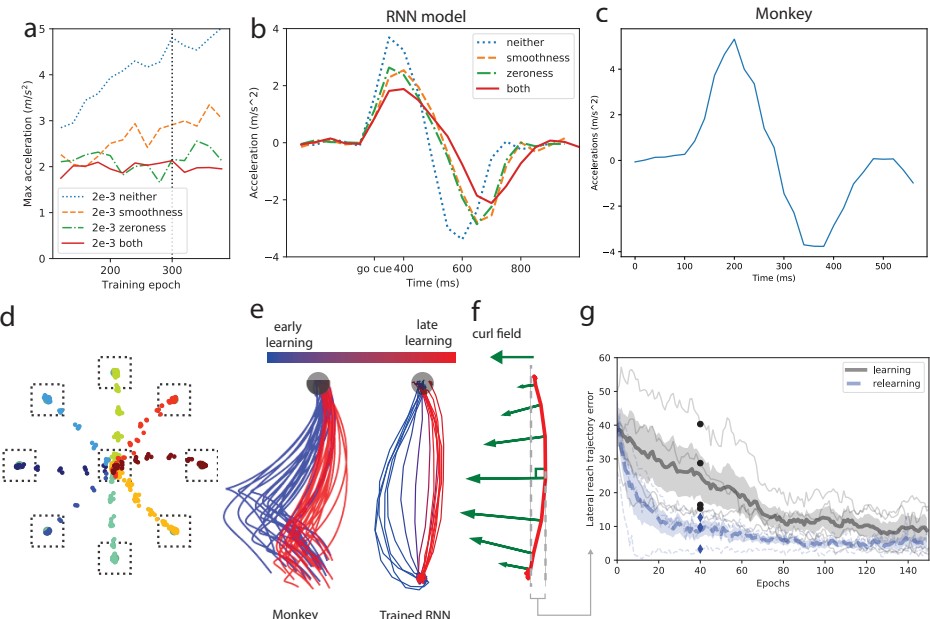

Figure 2: **(a)** Maximum acceleration during a reach as a function of pretraining epoch. Note that both models that had a zeroness regularization saturate around $2\ m/s^2$. **(b)** Acceleration profiles for reaches at pretraining epoch 300. Note that without regularizations, the reach completes faster, at the cost of increasingly unrealistic accelerations. **(c)** Acceleration profiles for monkey hand. Note the qualitative resemblance to the RNN acceleration profiles. **(d)** Example model reaches after 400 epochs of training. **(e)** The early and late curl field learning reaches of a monkey from Sun et al. (2022) (left) and our trained LSTM model (right). **(f)** An example reach at the end of the learning block, showing the curl field force vector for each time step. The apparently "overcompensated" reach is physically optimal, as it takes advantage of the curl field to reach faster. **(g)** Relearning, in blue, progressed faster than learning, in grey, replicating human and monkey behavior. (p¡0.05, Wilcoxon rank-sum test, one-sided

distribution constraints, the RNN model exhibited early and late learning reaching trajectories that are similar to the monkey's (Fig. 2e, left). In early learning (0-20 epochs), we observed that model reach trajectories were perturbed to the left by the clockwise CF (force applied left for a downward reach, blue trajectories in Fig. 2e). Over the course of learning, we observed these trajectories deviate less from the goal target (red trajectories in Fig. 2e). These results indicate that models learned to reach in the CF only through a sparse reward (+1 for successful target acquisition).

After learning to reach to the "down" target in the curl field environment, the model exhibits a curved reach, "overcompensating" for the curl field (Fig 2e, green arrows), a behavior also observed in humans (Izawa, 2008). This apparent overcompensation is optimal under the curl field, because it takes advantage of the curl field to boost acceleration or deceleration in the appropriate half of the reach. This behavior emerges naturally out of the reward maximization inherent to PPO, but would have to be analytically calculated, then enforced in through supervision, had we took a supervised learning modeling approach. While the trajectories aren't exactly the same (Fig. 2e, e.g., the RL model executes straighter reaches than the monkey), the RNN models capture the key hallmarks of successfully learning behavior.

**RL models exhibit motor memory:** A hallmark of motor memory is that humans and animals, when asked to relearn a task they were previously exposed to, will learn the task more quickly in later exposures than the initial one (Krakauer et al., 2019). To test whether RNN models exhibited this kind of motor memory, we first performed full washout of the learned behavior. When the curl field was initially removed, RNN model trajectories deviated to the right, indicating motor memory of the curl field. After approximately 100 episodes of reaches without a curl field perturbation, the models performed relatively straight reaches in the washout task block, with little deviation from

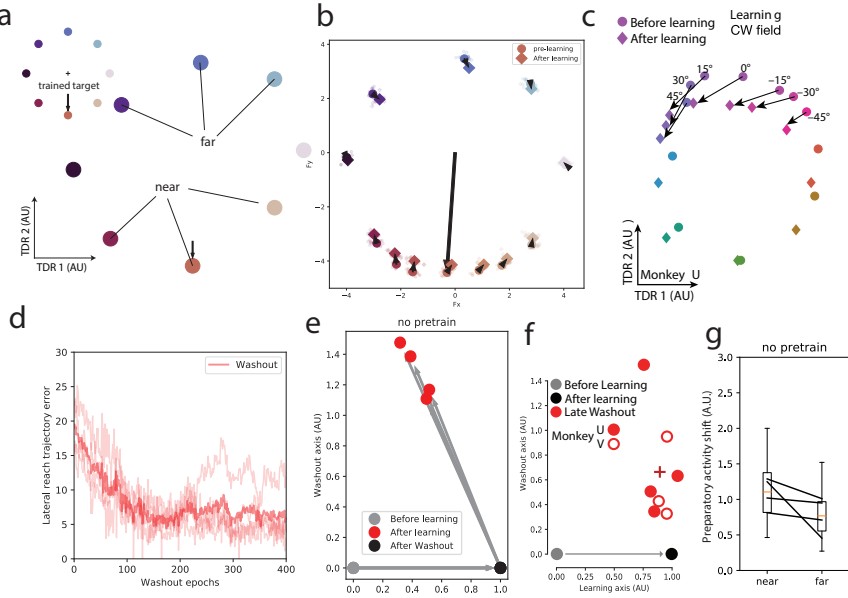

Figure 3: **(a)** Preparatory states before learning curl field. Inset is the reaching environment for the RNN model, with color coded targets. **(b)** RNN before-learning preparatory states and after-learning preparatory states, both visualized in the force-predictive subspace identified by TDR before the learning phase. Preparatory state corresponding to the trained target indicated by large black arrow. **(c)** Monkey before-learning preparatory states and after-learning preparatory states, both visualized in the force-predictive subspace identified by TDR before the learning phase. **(d)** Lateral hand deviation during washout block of adaptation experiment. Average across 4 seeds in bold red. **(e)** Learning (black) and washout (red) states plotted on the learning-washout plane. **(f)** Learning and washout shifts for monkeys, plotted the same way, from Sun et al. (2022). **(g)** Average shift magnitude along the learning axis for targets near or far from (see (a)) the trained target. Shifts are not statistically different (p=0.08, Wilcoxon rank-sum test, one-sided), although near shifts are consistently larger than far shifts.

a straight reach (Fig. 3d), washing out the learned curl field behavior. We then exposed the model to the curl-field environment again. Despite full washout of the learned behavior, we observed relearning was faster than learning (Fig.2g). During re-learning, the RNN policy achieved a lateral reach trajectory error on the order of 10 mm within 20 epochs, where as this took approximately 100 epochs during initial learning. We emphasize that the learning phase is preceded by a pretraining phase, where the model can already perform unperturbed reaches, and it is followed by a complete washout period, where all curl field adaptations are unlearned. RNN models therefore relearned the CF task more quickly than initial learning, despite full washout.

Together, these results demonstrate that RL models replicate key behavioral characteristics present in animals and humans during motor learning.

### 3.3 RNNs TRAINED WITH RL REPLICATE SEVERAL NEUROPHYSIOLOGICAL CHARACTERISTICS

Since the RNN model replicated many of the behavioral characteristics of monkeys engaging in motor adaptation, we investigated the similarity between RNN model preparatory activity and biological neural activity over the different blocks of learning. We characterized the RNN model's "neural geometry" by the following 4 neurophysiological features observed in monkeys: (1) preparatory state geometry, (2) stability of the force-predictive subspace, (3) uniformness of preparatory activity shifts after CF learning, and (4) orthogonality of learning-washout shifts.

**Preparatory states:** It is well known that neurons in the motor cortex fire in preparation for a reach to a target (Sun et al., 2022; Churchland et al., 2006b;a; Santhanam et al., 2006; Churchland

et al., 2010). When neural activity during movement preparation is collectively visualized in a 2D projection (found via PCA, factor analysis, or TDR (Mante et al., 2013)), they are organized according to the topology of the physical locations of the reaching targets (Santhanam et al., 2009). We visualized the preparatory activity in the RNN model by first constructing time series of the artificial neuron activations during the delay period of the task. We then performed dimensionality reduction using TDR (Mante et al., 2013). We identified two orthogonal dimensions where neural activity predicts cursor acceleration and projected the preparatory activity into these dimensions. We found that the models, across all 4 seeds, possess preparatory states (Fig. 3b) and that the preparatory states are organized in a circle, mirroring the geometry of the targets. These results therefore demonstrate that RL-trained RNN models exhibit similar preparatory state geometry to trained animals.

**Stable force predictive subpsace**  One of the discoveries of Sun et al. (2022) is that the force-predictive subspace found by TDR remains force-predictive throughout subsequent phases of learning, washout, and relearning. The stability of this subspace across learning for the monkey can be seen in Fig. 3c (see Fig.1 for all seeds), where the before and after learning preparatory states for any given target remains stable in the force-predictive subspace. Likewise, Fig. 3b shows that the preparatory states remain largely stable in this force-predictice subspace found before learning for the RNN as well.

**Uniform preparatory state shifts**  The most surprising result of Sun et al. (2022) is that even though the CF is behaviorally trained for one target (the down target), the preparatory states of all eight targets shift, reflecting newly learned motor cortex neural population states to perform the task. Further, this shift is not larger for the trained target, but uniform across all targets. In sum, Sun et al. (2022) reported that preparatory activity evolves over adaptation outside of the force (acceleration)-predictive subspace, with all eight preparatory states shifting by approximately the same distance and direction.

We performed this analysis on RNN models after they were also trained to reach through the curl field towards the down target. We first computed the learning axis, which is the vector that points from the average of the preparatory states before learning to the average preparatory state after learning. Projecting the preparatory states onto this learning axis, we then compared the average shift of the three targets near the trained target (down, down left, and down right targets, Fig. 3a), compared to the three far targets (up, up left, and up right targets, Fig. 3a). Surprisingly, even in RNN models, we found that the shifts are uniform for both pretraining conditions (Fig. 3f, shifts not significantly different, $p > 0.05$, Wilcoxon rank sum test), reproducing the findings from neurophysiological experiments.

Together, these results demonstrate that RL-trained RNN models, with no direct supervision, exhibit a key hallmark of cortical motor learning: a uniform shift in preparatory state geometry across all targets during CF learning, even though the CF was applied to only one target.

### 3.3.1 HYPOTHESIS: PRIOR EXPERIENCE INCREASES SIMILARITY TO BRAIN NEURAL GEOMETRY

The last measure of brain-like neural geometry was orthogonality of the learning-washout shifts.

**Nearly orthogonal learning and washout shifts**  During CF washout, Sun et al. (2022) quantified a washout axis by substracting the mean preparatory state after washout with the mean preparatory state after learning. The before learning, after learning, and after washout states can then be viewed in the 2D space spanned by the learning and washout axes. For monkeys, the learning and washout shifts were found to be close-to-orthogonal in this space across days (Sun et al., 2022) (Fig 3e). This critically shows that washout is not just an undoing of the uniform shifts, but results in entirely new preparatory states that are distinct from preparatory states before learning, in spite of identical behavior. It is thought that these new preparatory states represent a motor memory of the CF, enabling faster relearning (also observed in our RNN models in Fig. 2g). We therefore quantified if RNN preparatory states after washout returned to pre-learning states or new states, and whether these shifts were orthogonal to the learning axis.

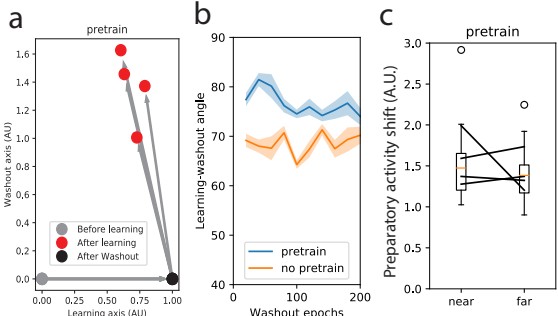

Figure 4: **(a)** Learning (black) and washout (red) states plotted on the plane spanned by the learning and washout shift vectors, for the CF pretrained model. **(b)** Average angle between the washout shift (black to red dot, panel **(a)**), and learning shift (grey to black dot, panel **(a)**), for both the CF-pretrained and baseline model, across 200 epochs of washout. Noticeably, prior experiences causes the learning-washout shifts to become significantly more orthogonal in the learning-washout shift plane, indicating an increased match to neurophysiological data. Shaded area is s.e.m. over 4 seeds. **(c)** For the CF-pretrained model, shifts are found to be uniform (p=0.38, Wilcoxon rank-sum (one-sided)), like the baseline model, maintaining the reproduction of this aspect of macaque neural data.

We observed that while the RNN model exhibited washout shifts that did not return to before learning states, the shifts also significantly regressed on the learning axis (Fig. 3e). These models therefore did not replicate a key neurophysiological finding. We hypothesized this may be because these networks have significantly less prior movement experience than monkeys, and were thus not endowed with general purpose dynamical motifs (Sussillo et al., 2015; Driscoll et al., 2022) that could be leveraged to reach under a CF. That is, these networks had to learn how to reach through a CF *de novo*, unlike monkeys that have a lifetime of prior experience.

We hypothesized that if the RNN model was pretrained to reach under curl fields, it would develop more structured dynamical motifs corresponding to various curl field intensities; The model would then exhibit more brain-like neural geometry during learning and washout as a consequence of leveraging (Driscoll et al., 2022) the additional recurrent circuitry created from richer prior experience.

**Pretraining with prior experience with curl fields**    We thus constructed another pretraining procedure (CF-pretrained RNN model) that differs from the original procedure in the following ways

1. Pretraining occurs with a random curl field perturbation. The curl field coefficient $k$ from Eq. 1 is set to a random value from $(-K, K)$ after every successful reach, where $K$ grows from 0 to 0.4 in steps of 0.01 with the curriculum.

2. The curl field coefficient $k$ is also provided to the network. We provide $k$ to the network to encourage the network to develop structured dynamical motifs that can deal with a range of curl fields. In nature, a monkey moves their hands under a range of dynamics. For example, a monkey may throw rocks. By sight, the monkey can estimate the weight of the rock. Internally, it thus has an estimate of the parameters characterizing the dynamics of the environment, before any physical interaction. The monkey likely arrived to the experimental room with a vast range of such prior experience, where the underlying parameters characterizing environment dynamics are observed. We reason that providing the curl field coefficient to the model imitates this sort of prior experience.

After the pretraining, the model input corresponding to the curl field coefficient is zeroed out. The CF-pretrained RNN model does not differ from the baseline model in input after pretraining.

We quantified preparatory state shifts during learning and washout for RNN models with CF-pretraining (Fig. 4a). We observed that they closely replicate close-to-orthogonal shifts also observed in Sun et al. (2022) (Fig. 3e). We emphasize that CF pretraining replicated these results with no explicit additional constraints on model weights or model activity, resulting in washout shifts that were more orthogonal to the learning shifts (Fig. 4b). As shown in Fig. 4b, we found that the

mean angle between the learning and washout shift was significantly closer to orthogonal for the CF pretrained model, compared to the model without prior experience with the curl field.

The added prior experience has not detrimentally affected any of the other key features of neural activity. The preparatory states are still organized circularly, and the force-predictive subspace remains stable (Fig.2). The preparatory state shifts remain uniform (Fig 4c). Together, these results demonstrate that RNN models exhibit many neurophysiological features of biological motor memory, especially when given adequate prior experience.

## 4 DISCUSSION

We found that our KL-divergence based policy regularization terms could induce realistic reach trajectories in our artifical motor adaptation experiment. We then showed that both of our models naturally replicated an array of behavioral findings from motor adaptation Sun et al. (2022), including overcompensated reaches in a curl field and evidence of motor memory through increased relearning speed. In addition to behavioral replication, we also found activity resembling neurophysiological recordings from monkey motor cortices Sun et al. (2022). In particular, the preparatory states exist in a stable force-predictive subspace, and shifted uniformly over the course of learning.

Finally, we explored the impact that prior experience has on the activity underlying adaptation to a curl field. We found that CF-pretrained models replicated the more orthogonal relationship between learning and washout shifts observed in monkey experiments. From this, we hypothesize prior experience may have induced the development of structured dynamical motifs Driscoll et al. (2022)that then shape the neural geometry observed through learning and washout (Sun et al., 2022). Future work should also seek to characterize properties of the weight changes underlying the activity changes.

Our findings around pretraining parallel suggestions from a number of researchers that accurate models for cognition may require pretraining (Yang & Molano-Mazón, 2021). Humans, for example, enter each situation with significant prior knowledge. Similarly, before beginning the motor adaptation experiment (Sun et al., 2022), the monkeys had developed motor skill in a wide array of tasks, and their motor cortices likely encoded significant priors about the relationships between different magnitudes and types of perturbing forces, as well as generated movements.

Future work ought replicate additional neurophysiological phenomena. Sun et al. (2022) characterized relationships between learning shifts of different trained targets, or opposite curl fields, as well as a "retrieval" of motor memory, where relearning caused the preparatory states to return to the learning state. While behaviorally, we found that relearning is faster than learning, indicating some form of motor memory, our RNN models may not be using the same underlying mechanisms.

Future work may further investigate the algorithms, objective functions, architectures, or task-set priors that implement motor memory in the brain. It may be interesting to investigate the impacts of continual learning algorithms inspired by neurobiological observations (Zenke et al., 2017; Kirkpatrick et al., 2017) on model activity, to expand the model's pretraining set to contain multiple tasks (Yang & Wang, 2020), or more ecologically relevant tasks (Molano-Mazón et al., 2023). Additionally, it may be interesting to investigate the impact of reward shaping, curriculum learning, or different RNN architectures on learning-related model activity.

Our work involved simple comparisons of angles or distances in neural space. Future work could also develop or include additional methods of specifying salient or important features of neural activity, and quantifying the similarity between these features in artificial models and animal models.

## 5 CONCLUSION

We have demonstrated that artificial models can learn curl field reaching in a way that reproduces several key behavioral and neural features in animal learning. Our method constrains motor adaptation behavior by using action distribution regularization terms while training RNNs with PPO. Our models, regardless of prior experience, replicated key behavioral findings in monkey motor adaptation experiments, and several neurophysiological features. We also show that pretraining with randomized curl fields leads to an increase in brain-like neural geometry, suggesting that previous

motor memories influenced the neural geometry observed in Sun et al. (2022). Previous work training RNNs with RL for neuroscience was limited to cognitive and value-based tasks with discrete action spaces (Song et al., 2017) or reaching tasks without consideration for the learning process (Marin Vargas et al., 2024). To the best of our knowledge, our work is the first that investigates the similarity between RL-trained RNN model activity during learning to the monkey motor cortex during motor learning, and puts forward the hypothesis that near orthogonal aspects of neural geometry underlying learning and washout may be a product of structured dynamical motifs formed through prior experience.

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

# A APPENDIX

## A.1 KL DIVERGENCE REGULARIZATIONS

Since the action distribution is modeled as a 2D diagonal Gaussian representing the acceleration in Cartesian coordinates, the KL divergence terms can be decoupled into its x and y components. We therefore perform the derivation of the KL divergence between two univariate Gaussian distributions below. The derivations generally hold for each component of the 2D Guassian output distribution, so we will omit any subscripts referring specifically to the x or y component.

The KL divergence between two univariate Gaussian distributions can be derived and found to be

$$
\begin{aligned}
KL(p,q) &= -\int p(x) \log q(x) dx + \int p(x) \log p(x) dx \\
&= \left[ \frac{1}{2} \log(2\pi\sigma_2^2) + \frac{\sigma_1^2 + (\mu_1 - \mu_2)^2}{2\sigma_2^2} \right] - \left[ \frac{1}{2} \left( 1 + \log 2\pi\sigma_1^2 \right) \right] \\
&= \log \frac{\sigma_2}{\sigma_1} + \frac{\sigma_1^2 + (\mu_1 - \mu_2)^2}{2\sigma_2^2} - \frac{1}{2}.
\end{aligned}
$$

For the smoothness constraint, this reduces to

$$
KL(\pi_\theta(\cdot|s_t)||\pi_\theta(\cdot|s_{t+1})) = \log \left( \frac{\sigma_{t+1}}{\sigma_t} \right) + \frac{\sigma_t^2 - \sigma_{t+1}^2}{2\sigma_{t+1}^2} + \frac{(\mu_t - \mu_{t+1})^2}{2\sigma_{t+1}^2}
$$

The first two terms of the smoothness constraint serve to ensure that the output acceleration standard deviation does not change too rapidly from timestep to timestep. The last term performs a L2 regularization on the change in acceleration in a timestep, per component. This insures that the "jerk" is low, and relates to the "angle jerk" cost function found to influence the shape of all reaches performed by humans in Berret et al. (2011).

For the zeroness constraint, this reduces to

$$
KL(\pi_\theta(\cdot|s_t)||\mathcal{N}(0,1)) = -\log \sigma + \frac{\sigma^2 - 1}{2} + \frac{\mu^2}{2} \tag{4}
$$

The first two terms in the zeroness constraint serve to constrain the variance near 1, which is found to be helpful for reliable convergence (Appendix Fig. 8). The last term acts as a L2 constraint on the model's "hand" acceleration, which is related to the "angle acceleration" cost function found to influence the shape of all reaches performed by humans in (Berret et al., 2011).

Taken together, our behavioral regularizations have empirical backing in human experiments, as they are related to the regularizations on joint acceleration and jerk that are identified through inverse optimal control in humans (Berret et al., 2011).

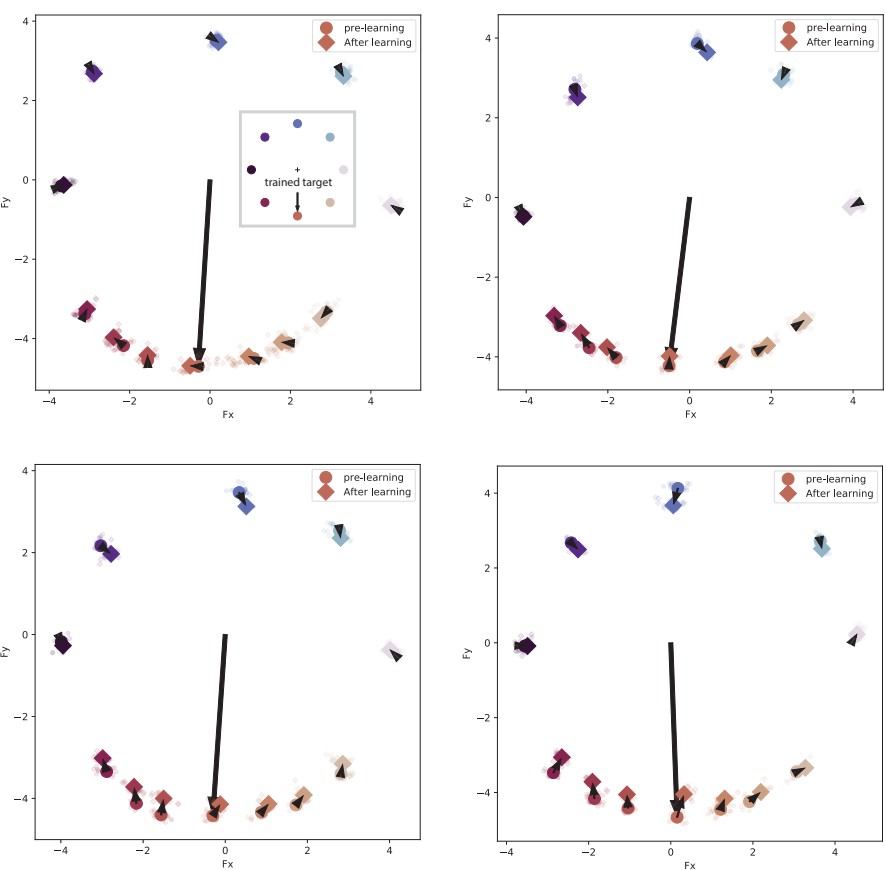

Appendix Figure 1: (TDR, no prior experience) Preparatory states, before and after learning, for the model *pretrained only on unperturbed reaches*, for each of the 4 seeds. These preparatory states are visualized in the 2-dimensional force-predictive subspace found by performing TDR on the before-learning preparatory states. This demonstrates that the force-predictive subspace remains stable across the learning phase.

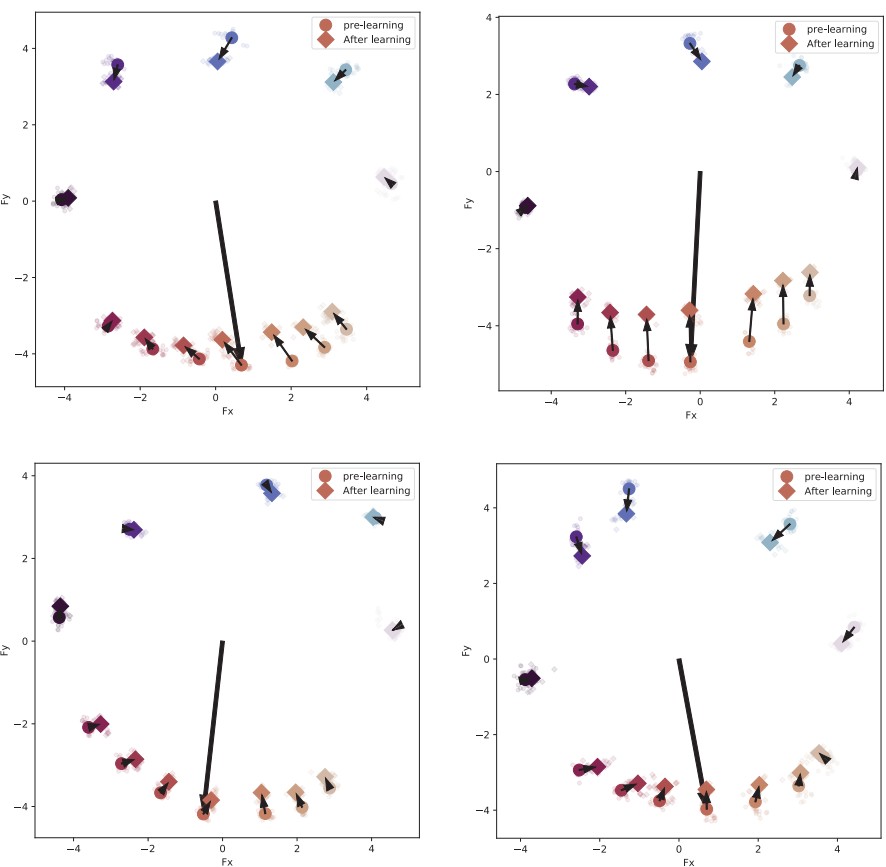

Appendix Figure 2: (TDR, prior experience) Preparatory states, before and after learning, for the model *pretrained to perform curl-field perturbed reaches*, with observation of the curl-field coefficient (during pretraining only), for each of the 4 seeds. These preparatory states are visualized in the 2-dimensional force-predictive subspace found by performing TDR on the before-learning preparatory states. This demonstrates that the force-predictive subspace remains stable across the learning phase.

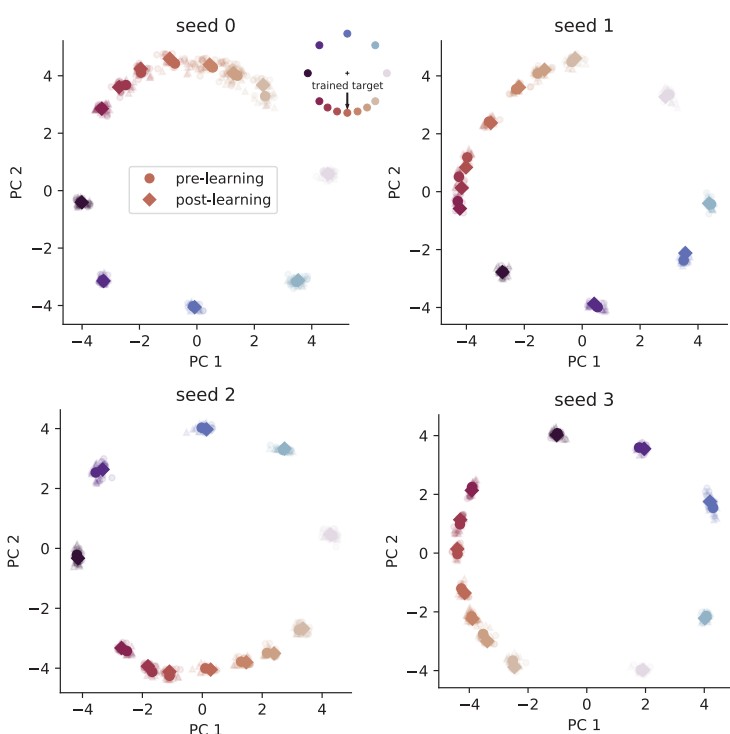

Appendix Figure 3: (PCA, no prior experience) Preparatory states, before and after learning, for the model *pretrained only on unperturbed reaches*, for each of the 4 seeds. These preparatory states are visualized in the top 2 principal components. This demonstrates that the force-predictive subspace lies largely in the top 2 variance dimensions, and remains stable across the learning phase.

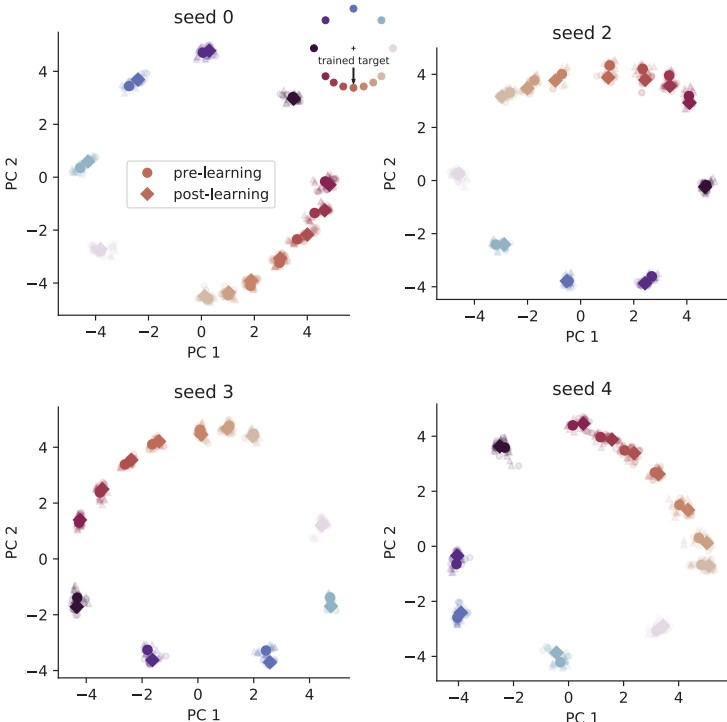

Appendix Figure 4: (PCA, prior experience) Preparatory states, before and after learning, for the model *pretrained to perform curl-field perturbed reaches*, with observation of the curl-field coefficient (during pretraining only), for each of the 4 seeds. These preparatory states are visualized in the top 2 principal components. This demonstrates that the force-predictive subspace lies largely in the top 2 variance dimensions, and remains stable across the learning phase.

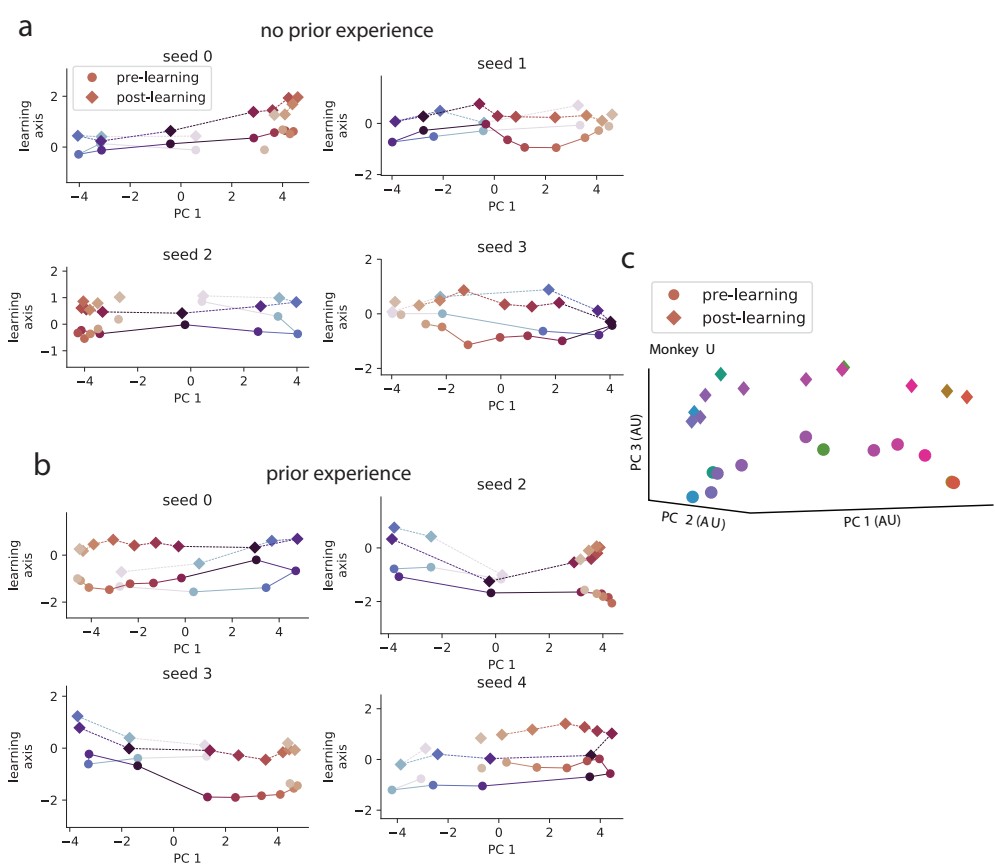

Appendix Figure 5:    (a, b) Preparatory states before and after learning, visualized on the first principal component and the learning axis (orthogonalized) for (a) models with no prior experience with curl fields before the experimet and for (b) models with prior experience with curl fields prior to the experiment. (c) Monkey preparatory states before and after learning visualized in the top 3 principal components reveals a uniform shift in the 3rd PC component. We visualized our data along the learning axis instead of the 3rd principal component because the model's preparatory state ring exhibited high curvature that occupies the third PC, causing the shift to be in subsequent PCs.

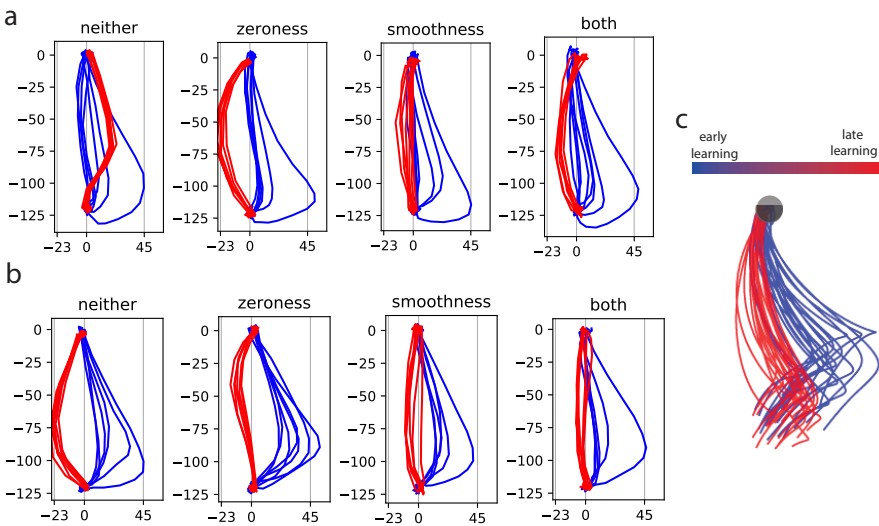

Appendix Figure 6: (a, b) Model reaches towards the down target during early (blue) and late (red) learning, for two seeds. With neither constraints, reaches do not reliably converge to even reasonable behavior (top, neither). With the zeroness constraint, the model leads to overcompensate for the curl field in order to utilize it to maximize reaching speed and hence reward, but at the cost of match to monkey reaches (see c). The smoothness constraint causes the reaches to be straight, and in combination with the zeroness constraint we obtain more reliable and reach trajectories most like the monkey's. (c) Monkey reaches towards the down target during early and late learning, for comparison.

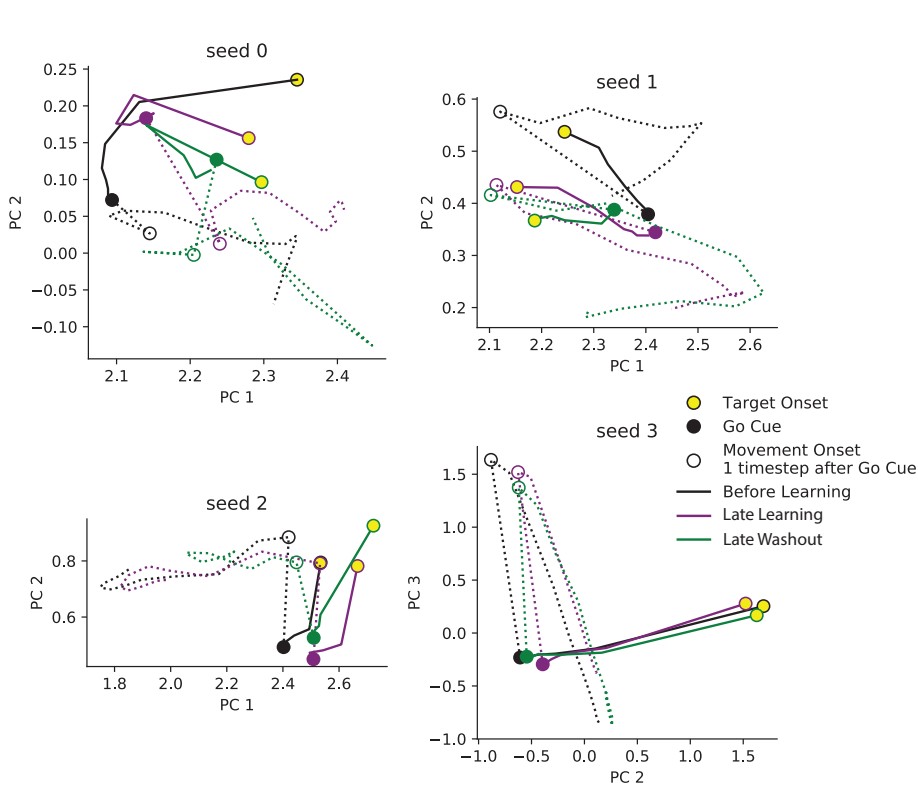

Appendix Figure 7: (no prior experience) Neural trajectories for before-learning, late-learning, late-washout models that are not exposed to the curl field prior to learning it. Movement preparation is indicated in solid lines, and movement execution is indicated in dotted lines. The trajectories proceed from the yellow centered cirles (target onset) through solid lines (movement preparation) to the solid circles (end of delay period), where the go cue is turned on. The model then takes one timestep to initiate movement (movement initiated during the empty circles), and proceeds to execute the movement (dotted lines). Notably, through learning and washout, the neural trajectories maintain significant resemblance, despite $\sim 10^8$ timesteps of training.

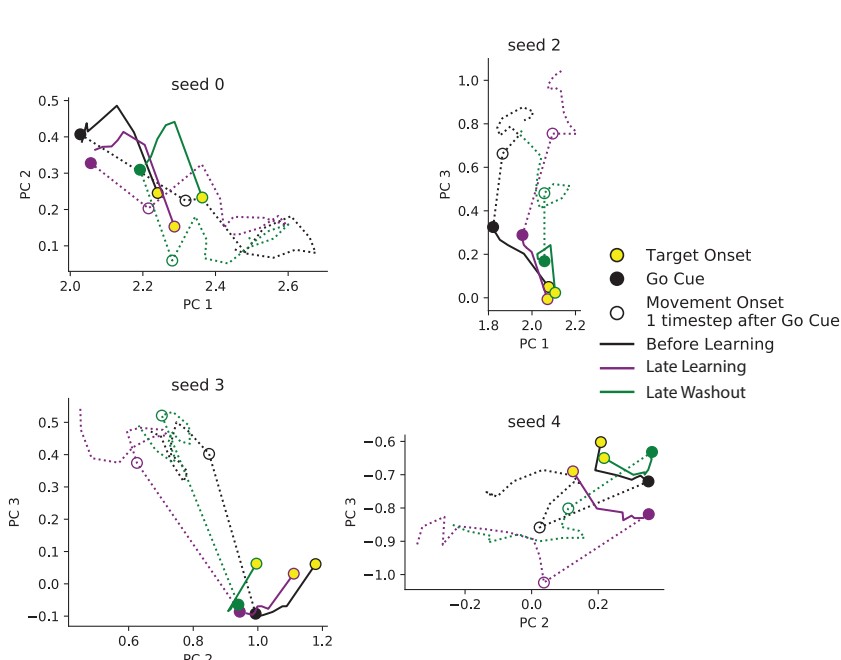

Appendix Figure 8: (prior experience) Neural trajectory for before-learning, late-learning, late-washout models that *are* exposed to the curl field prior to learning it in the experiment. Movement preparation is indicated in solid lines, and movement execution is indicated in dotted lines. The trajectories proceed from the yellow centered cirles (target onset) through solid lines (movement preparation) to the solid circles (end of delay period), where the go cue is turned on. The model then takes one timestep to initiate movement (movement initiated during the empty circles), and proceeds to execute the movement (dotted lines). Notably, through learning and washout, the neural trajectories maintain significant resemblance, despite $\sim 10^8$ timesteps of training.

