# OpenReview forum: "Using Reinforcement Learning to Investigate Neural Dynamics During Motor Learning"
_ICLR.cc/2025/Conference — Submitted to ICLR 2025_

### Official Review · Reviewer_L1fe · 2024-10-31

**Soundness:** 4
**Presentation:** 3
**Contribution:** 4
**Rating:** 8
**Confidence:** 4

**Summary:**

To understand the development of animal motor learning, the authors seek to match the behavioral and neural activity found in prior work within a recurrent neural network. Using reinforcement learning, the authors demonstrated that artificial models ability to perform in an adaptation task in a manner similar to that seen previously in animal models. By replicating previously behavioral work within simulation, the authors use these models to provide evidence that the learning-washout neural geometry underlying motor memory is influenced by prior neural representations.

**Strengths:**

A notable strength of this work comes from the authors' commitment to biological relevance over computational convenience, despite submitting to a computational conference. The intentional consideration of ecological validity sets this work apart; in replicating Sun et al.'s study, their model demonstrated emergent neural-like dynamics during motor adaptation, rather than being explicitly designed to simulate biological patterns. Particularly interesting was their finding regarding orthogonal shifts in memory storage - these patterns only emerged after pretraining. While the pretraining had minimal impact on reaching behavior, it led to activity patterns that more closely matched neurophysiological observations, suggesting the model naturally developed biologically relevant computational strategies.

**Weaknesses:**

The conclusions focus more on the neural dynamics of motor learning but most of the interpretations focus on behavioral performance. I suggest that the authors expand on the discussion with a deeper interpretation of the weight changes in the network, rather than focusing primarily on the behavioral performance.

**Questions:**

Outside of behavioral performance, how would you interpret the RNN neural weight development in relation to animal models?

---

> ### Author Response · Authors · 2024-11-25
>
> Thank you for the review! Investigation of the properties of weight changes that reflect more or less brain-like learning activity is indeed interesting. We are excited to explore this in future work. We will expand the discussion to comment on this.

---

### Official Review · Reviewer_9Hh8 · 2024-11-03

**Soundness:** 3
**Presentation:** 2
**Contribution:** 3
**Rating:** 5
**Confidence:** 4

**Summary:**

This paper use recurrent neural networks (RNNs) model to reproduce observations in a motor learning task with curl field adaptation.  The model is shown to replicate key behavioral and neurophysiological features observed in monkey experiments, such as faster relearning rates and stable force-predictive subspaces. It also showed that the inclusion of prior curl field experience enhances brain-like neural geometry in learning-washout shift orthogonality, supporting the hypothesis that structured dynamical motifs contribute to motor memory.

**Strengths:**

This work applies reinforcement learning to model motor learning dynamics, rather than supervised learning which are usually applied.

The inclusion of new regulation terms in the objective function facilitate the reproduction of experimental observation (but see following).

The modeling is well-motivated and rigorous, providing a significant step forward for understanding motor adaption (but see following).

**Weaknesses:**

1. The inclusion of two regularization terms in the objective function of PPO helps to yield realistic trajectories. These two terms, representing smoothness and zeroness constraints, is motivated by the paper Berret et al. 2011. But how the 8 cost terms in the cited paper relating to the two proposed terms formulated as the divergence of policy needs more explanation and elucidation.

From Fig.2a,b,c,  it seems that without the two additional terms in the objective function, the model performs ok when comparing the acceleration from the model to the monkey behavior. Therefore it need more justification about the inclusion of the two terms.

2. This work mainly reproduces the observation in Sun et al. 2022 paper. In that paper, the neural population trajectories in the PCA space were also presented for motor learning and washout. This was not examined in the current modeling work. A comparison of the population activity for trained RNN and those from monkey experiments will provide more information and justification of the model.

Some variants of the tasks that applied in monkey experiments are not examined in the RNN modeling. For example, in the current paper, the curl field is applied only to one target. But in monkey experiments, the sequentially different or interfering curl field was also investigated.

3. The environment in the model is based on gym and the algorithm was an extension of PPO. But some details are not clear from the method. For example, the input dim is denoted either 9D (Line 145) or 10D (Line 183). It would be much better to provide the code for evaluation and reproducibility.

**Questions:**

Could the authors provide more details on how the smoothness and zeroness constraints align with the cost functions in Berret et al. 2011?

Could the authors provide comparison of population activity from model and monkey experiment?

Could the model reproduce observation in the task variants including sequentially different or interfering curl field as in monkey experiments?

There are some improvements about text and figures are needed:
1. In Eq. 3, the two new added terms miss the expectation as in the PPO object function.
2. Line 14: "We sought train", Line 431 "while while", Line 426: "inspite of"; Line 484: "a that"; L485: "closer to orthogonal”.
3. Some figures are not well illustrated. For example, in Fig.4b, lines extend outside of the figure boundary. In Fig.3g, no label for y axis.

---

> ### Author Response · Authors · 2024-11-25
> **Rebuttal**
>
> Thank you for your review, thoughtful questions, and suggestions. We reply to them in order.
>
> >how the 8 cost terms in [Berret et al. 2011] relate to the two proposed terms formulated as the divergence of policy needs more explanation and elucidation.
>
> **Connection to Berret et al., 2011**: Berret et. al. 2011 find that of 8 cost terms, 3 terms make up a majority of the cost function:
> $$
> C_7 = \int_0^T  |\dot{\theta}_1 \tau_1| ...dt
> $$
> $$
> C_2 = \int_0^T \dddot{\theta}_1^2 ... dt
> $$
> $$
> C_3 = \int_0^T \ddot{\theta}_1^2 ... dt
> $$
>
> $\theta$ corresponds to joint angles and $\tau$ to torque.
>
> Our KL divergence terms simplify (see Appendix) to:
>
> KL_zeroness
> $$
> KL(\pi_\theta(\cdot|s_t)||\mathcal{N}(0,1)) = -\log \sigma + \frac{\sigma^2 - 1}{2} + \frac{\mu^2}{2}
> $$
> KL_smoothness
>
> $$
> KL(\pi_\theta(\cdot|s_t)||\pi_\theta(\cdot|s_{t+1})) = \log \left( \frac{\sigma_{\text{t+1}}}{\sigma_t} \right) + \frac{\sigma_t^2 - \sigma_{\text{t+1}}^2}{2\sigma_{\text{t+1}}^2} + \frac{(\mu_t - \mu_{\text{t+1}})^2}{2\sigma_{\text{t+1}}^2}
> $$
>
> The KL_zeroness $\mu^2/2$ term is related to the squared angular acceleration cost by a linear model that is a function of the joint angles depending on the limb model. The smoothness term approximates the squared third derivative of position (jerk) and is thus related to the “angle jerk” term. Our model does not incorporate biomechanics; our primary goal is to analyze neural effects. We therefore directly apply this cost to the end effector’s acceleration. We will clarify this in the manuscript.
> >From Fig.2a,b,c, [without regularization] the model [... seems similar] to the monkey behavior. Therefore [the KL terms] need more justification
>
> Additional justification of KL terms: To address your concern, we performed additional analyses (Appendix Figure 6) to show these KL terms increase similarity to the animal’s trajectories. First, the trajectories with neither constraints typically fail to converge to a reasonable trajectory. Appendix Figure 6a show unconstrained trajectories that curve in the direction of the curl field after significant learning (red curves). This clearly does not match the monkey’s curl field reaches which curve in the opposite direction. Second, the overcompensation of the curl field becomes more consistent, but excessive, with the zeroness constraint alone. As described in Fig 2f, it is optimal to “overcompensate” in a curl field to speed up the reach (Fig 2f, lines 310-312). Smoothness-only constrained models take advantage of this mechanism and exhibit obvious overcompensation beyond what is observed in biophysical reaches (Appendix Figure 6c). Finally, the smoothness constraint helps to reduce curvature to better match the curvature of the monkey’s reach (Appendix Figure 6ab). Together, these constraints produce more consistent and monkey-like learning trajectories. We have added these results and clearer connections to Berret et al., 2011 to the manuscript.
>
> >neural population trajectories in the PCA space [...] for motor learning and washout [were] not examined in the current modeling work. A comparison of the population activity for trained RNN and those from monkey experiments will provide more information and justification of the model.
>
> Our initial submission included the neural population preparatory states, their stability over learning, and the structure of preparatory shifts over learning and washout (Figures 3a, b, e, g). To address your review for further comparisons, we added 2 figures of the PCA preparatory states across learning and washout (Appendix Figure 3 & 4) demonstrating stability in the force-predictive subspace, and one figure of the uniform shift along the “learning axis” and PC1 (Appendix Figure 5).
>
> >Could the model reproduce observation in the task variants including sequentially different or interfering curl field as in monkey experiments?
>
> We agree that these are interesting experiments. For our manuscript, we chose to focus on the behavioral constraints and replication of key neural and behavioral phenomena during motor learning over learning, unlearning, and relearning of a curl field. We believe this paper forms a foundation to build upon to reproduce other experimental observations in later work.
> >The environment in the model is based on gym and the algorithm was an extension of PPO. But some details are not clear from the method. For example, the input dim is denoted either 9D (Line 145) or 10D (Line 183). It would be much better to provide the code for evaluation and reproducibility.
>
> We apologize; the 10D was a typo. We will carefully correct any errors and clarify details in the Methods.  We will post the code with the camera-ready submission.
>
> Thank you for your comments and feedback. We fixed the typos you pointed out. We believe the new neural population activity figures will help clarify the relationship between the model and the monkey’s neural population activity. We hope these changes addressed your concerns.

---

> ### Comment · Reviewer_9Hh8 · 2024-11-28
>
> The authors have partially addressed my questions; however, two were only acknowledged without being fully addressed. The first pertains to reproducing neural population trajectories in the PCA space (Extended Fig. 9g in Sun et al., 2022, Nature), which serves as a valuable benchmark for comparing the trained RNN with neural population data. The second concerns comparing model predictions with task variants, such as sequentially different or interfering curl fields, as investigated in the monkey experiments in Sun et al., 2022, Nature. Both questions remain unresolved and require further clarification.

---

> > ### Author Response · Authors · 2024-11-29
> >
> > Thank you for the follow up. We have produced the trajectories of the RNNs (Appendix Figure 7 and 8) similar to Extended Data Figure 9. Like in Sun et al., we see that trajectories follow a curved trajectory. The washout trajectories (in purple) generally have dynamics that explore the state space in the vicinity of the pre-learning trajectory (black) and the learning trajectory (green). We will edit the camera-ready manuscript to remark on it. We hope this addresses your concern about the trajectories.
> >
> > For the second concern about task variants, we agree this is an interesting area of investigation – and our model is suited to test these questions – but we believe this investigation is outside the scope of this manuscript. We believe training RNNs to better replicate neurophysiology in sequential and interfering curl fields is an open area of future research, and may require additional innovations or training hypotheses. For example, we used pre-training (to model additional prior experience) as a way to induce more brain-like orthogonal shifts during acquisition and forgetting of one curl field environment in the RNNs; similar types of new training ideas, plasticity mechanisms, or regularizations may help induce more brain-like activity during sequential or interfering curl fields. While our model can test these questions, we believe that these open research directions is in the domain of future work, as our manuscript already covers significant ground: the first model of this task, as well as investigations into four phases of training (pre-learning, curl field learning, washout, and curl field re-learning) and how pre-training affects neural activity during these phases. We will make this clearer in the manuscript (lines 517-520).

---

### Official Review · Reviewer_Nys1 · 2024-11-04

**Soundness:** 3
**Presentation:** 3
**Contribution:** 3
**Rating:** 6
**Confidence:** 3

**Summary:**

The paper presented an experiment with an LSTM model trained with PPO on a classic motion learning task. In the task, a monkey moves a handle from one location to another location, while a disturb is applied during the motion. Detailed comparisons between the results of the model and the monkey were discussed.

**Strengths:**

It is interesting to compare the RL results with animals' learning results. To reproduce the classic experiment, the authors implemented a new RL environment, designed the reward, implemented models, and designed a learning curriculum. The comparison is in detail in various aspects, which are much better for scores only.

**Weaknesses:**

The proposed model is pretty simple, almost a direct reuse of existing models. The mathematical introduction of the model is not very clear, mainly because many variables were not explained, although a reader who is familiar with the RL field could guess what the author tried to show.

The comparison did not discuss the impact of curriculum learning, which might not be used in the experiment with a monkey.

The paper mentioned "brain-like neural geometry", while the evidence is not solid. At least a method to compare the geometry should be suggested and treat proof of "brain-like neural geometry" as a future work.

**Questions:**

1. Is there any model of the monkey's arm in the implemented environment? Or the agent just conceptially moving a handle?
2. Line 218, 2e4 is not a good practice for $2^4$ in a paper.
3. Line 233. No GPU is used? Is there any parallel computing with the CPU?
4. Figure 3. How these dots are located? Define the learning axis and washout axis.
5. Where are Figures S1 to S4?
6. What if the KL penalty terms were not used? A comparison with and without them can support the claim in Line 496 better.

**Details Of Ethics Concerns:**

Data from animal experiments

---

> ### Author Response · Authors · 2024-11-25
> **Rebuttal**
>
> Thank you for your review! We will address the weaknesses first, followed by the questions.
>
> >The proposed model is pretty simple, almost a direct reuse of existing models. The mathematical introduction of the model is not very clear, mainly because many variables were not explained [...]
>
> Thank you for this comment. We updated the methods section to clearly explain all variables. We will also clarify that, for the purposes of computational and systems neuroscience, we actually view model simplicity as a strength! It is desirable to have as simple a model that captures neural features as possible, since model complexities also complicate biological comparisons.
> The comparison did not discuss the impact of curriculum learning, which might not be used in the experiment with a monkey.
> We were empirically unable to train agents to perform this task without curriculum learning. We will state clearly that curriculum learning significantly facilitated training: A randomly initialized model struggled to learn delayed reaches without a curriculum if the acceptance window was too small, likely due to very sparse rewards.
>
> We wish to point out that, generally, to train monkeys to perform a delayed reach task, a curriculum is used. Naive monkeys initially train to perform delayed reach tasks with larger targets, and the target is progressively shrunk until it can perform a delayed reach task with a smaller target. Although there are differences, such as that the monkey due to prior experience can make physical delayed reach tasks and this training is therefore to cognitively teach him how to perform this, it does also point to the fact that a monkey is not born with the ability to perform precise and timed reaches based on cues, but learns it through experience. We will mention this in the manuscript.
>
> >The paper mentioned "brain-like neural geometry", while the evidence is not solid. At least a method to compare the geometry should be suggested and treat proof of "brain-like neural geometry" as a future work.
>
> With respect to comparing the geometry, we performed these comparisons (stated in Section 3.3): (1) preparatory state geometry, (2) stability of the force-predictive subspace, (3) uniformness of preparatory activity shifts after CF learning, and (4) orthogonality of learning-washout shifts.
>
> The first 2 are qualitative, and the last 2 are quantitative, in that they involve either finding statistically insignificant difference in shift for target near to or far from the trained target (3), or finding that the orthogonal shifts are statistically significantly different (4). Certainly, this can be abstracted and quantified further in future works. We have added such a suggestion to the discussion!
>
>
> Now we will address the questions. First, thank you for the style/typo suggestions. We have fixed the corresponding text in the updated manuscript.
> >What if the KL penalty terms were not used? A comparison with and without them can support the claim in Line 496 better.
>
> Please see our additional analyses (Appendix Figure 6) to show these KL terms increase similarity to the animal’s trajectories. First, the trajectories with neither constraints typically fail to converge to a reasonable trajectory. Appendix Fig. 6a (top left) shows exemplar learned trajectories that curve in the direction of the curl field after significant learning (red curves). This clearly does not match the biological behavior: the monkey’s curl field reaches curve in the opposite direction.  Second, the desirable overcompensation of the curl field becomes more consistent, but excessive, with the zeroness constraint alone. As described in Fig 2f, it is optimal to “overcompensate” in a curl field to speed up the reach. Smoothness-only constrained models take advantage of this mechanism and exhibit obvious overcompensation beyond what is observed in biophysical reaches (Appendix Figure 6c). Finally, the smoothness constraint helps to reduce curvature to better match the curvature of the monkey’s reach. Combining both constraints produces more consistent and monkey-like post-learning trajectories.
>
> >Is there any model of the monkey's arm in the implemented environment? Or the agent just conceptially moving a handle?
>
> There is no model of the monkey’s arm in our model. We only model the end effector kinematics.
>
> >Line 233. No GPU is used? Is there any parallel computing with the CPU?
>
> No GPU is used. Yes, we used vectorized environments, and ran 16 processes/environments in parallel.
> >Figure 3. How these dots are located? Define the learning axis and washout axis.
>
> The learning and washout axis definitions are in the text at lines 403 and 419. We will reorganized some of the text to make this definition easier to find.
> >Where are Figures S1 to S4?
>
> We have fixed the supplementary figures to start at S1 instead of S5. We have renamed them as Appendix Figures.

---

> > ### Comment · Reviewer_Nys1 · 2024-11-26
> >
> > How do you define neural geometry and how KL-divergence could explain the neuron mechanism of the learning?

---

> > > ### Author Response · Authors · 2024-11-27
> > >
> > > >How do you define neural geometry?
> > >
> > > We take the terminology used in Sun et. al. 2022, where they find that the “precise geometry of the uniform shifts may facilitate acquisition, retention, and retrieval of a broad motor repertoire”. In particular, they find that:
> > > 1. There is a stable force-predictive subspace, even as the monkey learns to generate different force profiles in response to different curl fields.
> > > 2. The uniform shift occurs in some direction orthogonal to the stable force-predictive subspace.
> > > 3. The uniform shifts during learning and washout have a “precise geometry”, in that they are orthogonal, therefore preserving the learning uniform shift as the behavior is unlearned (washed out).
> > >
> > > This gives a clear picture of how the preparatory states are related to each other geometrically in the neural activity state space. We used this notion of neural geometry of uniform shifts from Sun et al., 2022, and directly evaluated our artificial model for the same features. We will clarify in the paper that when we write “neural geometry,” it refers to these notions previously defined by Sun et al.
> > > >and how KL-divergence could explain the neuron mechanism of the learning?
> > >
> > > To be clear, the KL-divergences are only behaviorally – not neurally – motivated. They are used to induce monkey-like behavioral kinematics, and were not directly motivated by explaining the neuron mechanism of learning.  Please let us know if this does not clarify your question.

---

### Official Review · Reviewer_f5jW · 2024-11-19

**Soundness:** 2
**Presentation:** 3
**Contribution:** 2
**Rating:** 5
**Confidence:** 4

**Summary:**

This paper examines how training recurrent neural networks with reinforcement learning introduces the concept of motor memory, i.e., allowing curl field adaptation to affect future reaches. This is performed by introducing two regularization terms in a standard actor-critic reinforcement learning framework to train a network to produce acceleration that is then affected by curl-field forces. The authors then evaluate the network on their ability to replicate characteristics introduced in Sun et al., 2022, such as curved reaches, faster re-learning, and shifts in preparatory activity. The authors end with examining the effect of training with curl-field perturbations on the orthogonality of the learning to washout phases.

**Strengths:**

The paper examines the alignment between monkey neural experimental data and neural networks. How can we understand motor learning in computational networks? This paper sets up a reinforcement learning approach to the question. Understanding how network activity shifts during different epochs of a trial as well as more slowly during learning is a valuable goal.

**Weaknesses:**

1) There are very few baseline comparisons - which parts of the framework (loss / architecture / training) are necessary to see their main results? The paper makes reference to Richards et al., 2019, but does not adequately address this. The networks trained with reinforcement learning are not compared with networks that are trained with supervised learning, which is more common in this field.

2) While the resulting acceleration with the additional KL-regularization terms are shown, it is unclear which ones are more akin to the experiments since only a qualitative comparison is performed (and it seems like 'neither' is the most similar to the experimental data). In fact, the time axis is severely mismatched between the network and monkey data, and thus it is very difficult to compare the resulting traces.

3) The resulting curved traces in Figure 2e as in the trained RNN are not very similar to the monkey traces.

4) The difference between learning and relearning are not quantified adequately in Figure 2g. Is there a significant difference?

5) The effect of the curl field seems to be quite small in the networks in Figure 3b, as compared to the monkey in Figure 3c.

6) All the comparisons between the experiments and neural networks are qualitative; there is a lack of quantitative comparisons throughout the paper.

**Questions:**

While the authors examine the organization of preparatory states using TDR, previous experimental studies have also analyzed this using PCA, as cited by the authors. Does this structure only emerge using TDR or also PCA?

---

> ### Author Response · Authors · 2024-11-25
> **Rebuttal**
>
> Thank you for your review! We address the weaknesses and question below.
> >There are very few baseline comparisons - which parts of the framework (loss / architecture / training) are necessary to see their main results? The paper makes reference to Richards et al., 2019, but does not adequately address this. The networks trained with [RL] are not compared with networks that are trained with supervised learning, which is more common in this field.
>
> Regarding the loss and architecture, we will clearly state the RNN (rather than any feedforward architecture) is necessary because we are concerned about neural dynamics. We will also provide further detail that we were only successful training an LSTM (and were unable to train a vanilla RNN). For the reward, we directly mimicked the monkey’s juice reward for a successful target reach in order to maximize ecological validity. We recognize it is possible to perform reward shaping, but we view this as introducing unnecessary complexity for our goal of analyzing the effects of prior experience on neural activity. We will state that future work may explore the interesting question of how modifying rewards and even more complex RNN architectures may affect results.
>
> Regarding training, we are unaware of any other baseline for this motor learning task. Although supervised learning is appropriate for other tasks studying an already learned behavior, this task is unique in that it studies neural population activity during learning dynamics. In other studies in this field, we typically study the post-learning RNNs, but here, the difference in pre- and post-learning states are vital. Supervised learning would introduce many undesired complexities. For example, to match monkey learning, since the output is acceleration, we would need to define the entire target acceleration profile under a curl field. This is not realistic as the solution is known to the agent a priori through gradients that point to optimal behavior, rather than learning it through exposure (where we aim to study the emergent learning dynamics). It would also preclude analysis of degree of trajectory curvature under a curl field, since we would have already defined the final behavior supervision. This makes the supervised baseline unnatural for our task. We will further emphasize this in the manuscript.
>
> Finally, as it relates to Richards et al., we emphasize that their manuscript also stated “it will also be crucial to the development of a deep learning framework for systems neuroscience to focus on how a given animal might solve tasks in its appropriate Brain Set.” Our results serve as a piece of evidence that, with the loss, architecture, and learning rules fixed, the addition of prior experience (expansion of the Brain (task) Set to more than just movements performed or learned within an experiment, to those possibly seen during prior experience), can induce more brain-like learning activity.
> >unclear which [acceleration profiles] are more akin to the experiments since only a qualitative comparison is performed (and it seems like 'neither' is the most similar [...]
>
> To clarify the importance of the KL terms, we performed additional analyses (Appendix Figure 6) to show these KL terms increase similarity to the animal’s trajectories during learning of a curl field. First, the KL terms induce a better match to the monkey’s overcompensation behavior. As described in Fig 2f, it is optimal to “overcompensate” in a curl field to speed up the reach. Non-regularized models do not converge reliably. Smoothness-constraint models overly exploit this mechanism and exhibit overcompensation beyond biophysical reaches, while models constrained by both converge reliably while utilizing a level of overcompensation comparable to the monkey’s. We will provide quantitative statistics for this result in the camera-ready submission.
> difference between learning and relearning are not quantified adequately in Figure 2g. Is there a significant difference?
> We added S.E.M bars to Fig 2g. Relearning converges faster than learning at epoch 40 with (p<0.05).
> >All the comparisons [...] are qualitative; there is a lack of quantitative comparisons throughout the paper.
>
> In addition to the previous reply, we also included statistics for the uniform shifts (See Fig. 3g).
> >The effect of the curl field seems to be quite small in the networks in Figure 3b, as compared to the monkey in Figure 3c.
>
> The main neural effect of learning the curl field is apparent when examining preparatory state shifts outside of the PCA/TDR plane. This shift can be seen and compared to the shifts in monkey neural activity in the new Appendix Figure 5.
> >While the authors examine the organization of preparatory states using TDR, previous [...] studies have also analyzed this using PCA [...] Does this structure only emerge using TDR or also PCA?
>
> The structure of preparatory states is organized similarly in the PCA space, see our new Appendix Figures 3, 4, and 5.

---

### Meta-Review · Area_Chair_BPML · 2024-12-20

**Metareview:**

This paper explores how recurrent neural networks trained with reinforcement learning (RL) can be used to replicate findings from a monkey reaching task. The authors introduce regularization terms to an actor-critic RL framework, enabling networks to learn reaching movements and adapt to curl-field perturbations using RL alone. The paper demonstrates how this RL approach captures key aspects of motor learning, like curved reaches, rapid re-learning, and changes in preparatory activity, mirroring biological observations. The study also examines how training with curl fields affects the orthogonality of learning and washout phases, providing insights into motor adaptation.

This paper is primarily a contribution to computational neuroscience. There is no original computational method presented. Instead, the original finding is that RL can effectively model monkey reaching behavior, offering a more ecologically valid approach than previous supervised learning methods. This highlights the potential of RL+RNNs for explaining biological motor memory mechanisms.

Reviewers pointed out a lack of comparison to supervised learning baselines, which are common in this area, and questioned the strength of the fit to experimental behavioral data. They also pointed out missing task variants such as sequentially different or interfering curl fields from the original Sun et al, 2022 paper.

Since this submission is primarily a contribution to computational neuroscience, the missing task variants pointed out by reviewer 9Hh8 seem like a major missing piece for validating how well the presented model captures the phenomena from Sun et al, 2022. Given that there is no new computational model being proposed, and without a complete picture of a model for Sun et al, 2022, I do not believe this paper is suitable for ICLR.

**Additional Comments On Reviewer Discussion:**

Reviewer f5jW had several comments which were not easily addressed by the authors, but since the review was submitted late, it was discounted in the final decision. However, the reviewer made good points that the work could be strengthened by showing that the RL approach performs as well as a supervised baseline for this task (note, that even if it did not, it would still be useful as a baseline to show what RL+RNNs are still lacking as an explanation of this behavior).

Multiple reviewers mentioned running ablation experiments to verify the importance of the regularization terms, which the authors added to the revised manuscript during the rebuttal period.

Reviewer 9Hh8 raised concerns about missing task variants from Sun et al, 2022 which were not addressed during the rebuttal period.

---

### Decision · Program_Chairs · 2025-01-22

Reject